# Analysis of the demand for gastronomic tourism in Andalusia (Spain)

Mª Genoveva Dancausa Millán[1], Mª Genoveva Millán Vázquez de la Torre[2]*,
Ricardo Hernández Rojas[3]*

1 Department of Statistics, Cordoba University, Cordoba, Spain, 2 Department of Quantitative Methods,
Universidad Loyola Andalucía, Sevilla, Spain, 3 Department of Agrarian Economy, Cordoba University,
Cordoba, Spain

* gmillan@uloyola.es (MGMVT); ricardo.hernandez@uco.es (RHR)

## Abstract

In recent years, gastronomy has become a fundamental motivation to travel. Learning how to prepare gastronomic dishes and about the raw materials that compose them has attracted increasing numbers of tourists. In Andalusia (region of southern Spain), there are many quality products endorsed by Protected Designations of Origin, around which gastronomic routes have been created, some visited often (e.g., wine) and others remaining unknown (e.g., ham and oil). This study analyses the profile of gastronomic tourists in Andalusia to understand their motivations and estimates the demand for gastronomic tourism using seasonal autoregressive integrated moving average (SARIMA) models. The results obtained indicate that the gastronomic tourist in Andalusia is very satisfied with the places he/she visits and the gastronomy he/she savours. However, the demand for this tourist sector is very low and heterogeneous; while wine tourism is well established, tourism focusing on certain products, such as olive oil or ham, is practically non-existent. To obtain a homogeneous demand, synergies or pairings should be created between food products, e.g., wine-ham, oil-ham, etc., to attract a greater number of tourists and distinguish Andalusia as a gastronomic holiday destination.

**Data Availability Statement:** All relevant data are within the paper and its Supporting Information files.

**Funding:** The authors received no specific funding for this work.

## Introduction

Gastronomy constitutes an element of identity for a territory [1, 2], not only because of the raw material of the food product, e.g., oil, ham, cheese, wine, etc., in which the product is identified with the place of origin but also with the dishes that are created using them, which is a method of establishing a heritage that is inherited from one generation to another [3]. This heritage constitutes cultural and gastronomic wealth, and every day, interest in this tourism sector increases, not only because food serves to cover a basic need but also because food can be experiential; i.e., individuals are willing to "travel to eat" [4].

Gastronomy (raw materials, methods of preparing dishes and customs) can be considered part of the cultural identification of an area, providing its historical and cultural character and constituting, in cases such as "traditional Mexican cuisine", an Intangible Cultural Heritage, as recognized by UNESCO in 2010 [5], thus becoming a tourist attraction. Food has become a

**Competing interests:** The authors have declared that no competing interests exist.

tourist resource, and food products have increased the value of a destination [6]. The local identity can be determined by the territory and the food. Gastronomy, as a tourist resource [7], is appreciated not only for its own intrinsic value but also for its symbolic character, insofar as it acts as an identifier of towns and territories.

Andalusia, an region located in the south of Spain, has rich gastronomy comprising dishes whose recipes date back more than a thousand years, such as octopus casserole or eggplant almodrote, whose origin stems from the region's Jewish heritage. The latter is served at the Feast of Pesach or Passover and consists of a sauce made from oil, garlic and cheese that is added to vegetables and meats. The expulsion of the Jews from Spain in 1492 did not mean that the way of cooking in Andalusia was lost, and Judeo-Hispanic gastronomy developed widely in the Andalusian era, resulting in Spanish recipes that are still used and that have expanded throughout the Mediterranean and the East; these dishes were adapted to the local tastes of these places and created what is now called "Sephardic cuisine" [8–11]), in which olive oil is an essential element for cooking.

During the 800 years of coexistence of Jewish and Muslim people in the Iberian Peninsula, recipes such as Manchego gazpacho, which is prepared with unleavened bread (the absence of yeast; a direct inheritance of the Jewish tradition), Andalusian fish fry, Murcian meatloaf, Galician orejas de fraile (derived from the Orejas de Jamán), torrijas (Judeo-Spanish style French toast), Tarta de Santiago and desserts made with almonds were created. Because the precepts of the Kashrut prohibited yeast, there are dishes in which judo-Christian-Arab culinary traditions have been mixed, creating a fusion of gastronomic elements where culinary traditions are combined, giving rise to the current Spanish cuisine.

However, the methods of preparing dishes and the quality of the food elements that constitute typical dishes at a destination are now generating tourism, which is gaining more relevance every day and is becoming the main reason for tourists to visit certain areas [12]. Gastronomic tourists participate in some of the following activities: visits to primary and secondary food producers, gastronomic festivals, restaurants and specific places specializing in offering tastings of dishes and/or experiencing the attributes of a region that specializes in food production [13].

This study aims to analyse the profile of gastronomic tourists in Andalusia to determine what their motivation has been for completing the gastronomic route, what potential demand exists for gastronomic tourists and if demand is observed to be growing, to determine if it be an engine of development, i.e., economic growth, for the region; additionally the growth in demand is small or decreases, this study aims to determine whether promotional marketing campaigns can be used to improve it; the contribution of this research is the seasonal autoregressive integrated moving average (SARIMA) model of demand forecasting.

## Literature review and the protected designations of origin. PDOs

There is vast literature on gastronomic tourism, which has been analysed from different perspectives:

- Motivation: Motivation is presented as an important variable of tourism segmentation [14] and can be defined through three key issues in the tourism process, as follows: the reasons for the trip, the choice of destination and satisfaction [15]. However, it can be confirmed that as a general trend, the need to participate in an experience in which emotions play a relevant role seems to be present in all of them [16]. The resources offered by a country serve as a basis for this growing development; geographical and cultural diversity provides a great variety of foods and methods of preparing them [17]. The enhancement of these resources is providing new opportunities for many territories, especially rural territories, making this

type of tourism an important element of the economy and culture of these territories [18]. The development of gastronomic tourism contributes to integrating traditional primary productive functions with the specialized tertiary functions, increasing the sources of income and improving the income and employment levels of the local population and facilitating the multifunctionality of rural territories. However, it also favours cities; when gastronomy becomes an element that is part of the cultural of a city, tourists are inspired to make gastronomic journeys outside of the context of gastronomy festivals [19, 20]; rather, they desire an experience that provides heritage cooking [21], a hedonistic perspective [22], an opportunity to learn new dishes [23], a discovery of different culinary experiences [24] or the opportunity to eat at a restaurant included in a prestigious guide [25].

- Territory: Numerous studies have analysed the gastronomy of different locations. There are several studies that analyse gastronomic tourism in countries such as Turkey [26], Portugal [27], Japan [28, 29], China [30], India [31], Croatia [32], Greece [33–35], France [36–38], Hungary [39], Polonia [40], Serbia [41], Costa Rica [42], the United States [43], Russia [44, 45], Canada [46], Argentina [47], Ecuador [48] and South Korea [49], studies that address the original gastronomic practices performed in countries with emigration, such as Australia [50] and studies that explore the impact of European colonies on African gastronomy [51]. In Spain, local gastronomy has been studied in Córdoba [52] and Madrid [53].

- Raw materials: Studies regarding wine and oil are prominent [54, 55], as they are the main products that attract greater numbers of tourists, both nationally and internationally; however, other products, such as cheese [33], tuna, [56], cod [57], citron [58], insects [59], cephalopods [60], pizza [61, 62], wagyu [63], pork [64] and salmon [65], have been studied.

- Dish: Several studies have explored dishes prepared according to traditional recipes, such as salmorejo and oxtail [66], *plato minero* [67] and noodles [68].

Mediterranean Europe has vast culinary traditions, and quality raw materials are guaranteed by Protected Designations of Origin (PDOs) and Protected Geographical Indications (PGIs). In Spain, PDOs and PGIs provide a system for recognizing when a food product is of superior quality due to its special characteristics, which are unique due to the geographical environment in which the raw materials are produced and the products are processed and due to the influence of the humans who participate in the production.

For the empirical analysis, data on the existing active tourism companies were obtained from the official Andalusia Tourism Registry database (Regional Government of Andalusia) and the National Classification of Economic Activities (CNAE) for the period 2017–2018, in addition to economic data from the Trade Registry and, when needed, corporate and personal income tax returns and other reports from company websites (Table 1). All active, registered companies in that period were taken into account for the survey, resulting in a total of 44 companies. Companies that were active but not registered were not included in the survey sample (an additional 20%, approximately).

PDOs are the most well-known food quality distinctions. However, the Ministry of Environment and Rural and Marine Affairs of Spain grants other distinctions, such as the Qualified Designation of Origin (Denominación de Origen Calificada—DOCa), which must comply with, in addition to the requirements for appellations of origin, the following:

a. At least ten years has passed since its recognition as a PDO.

b. The entire product is marketed from companies registered and located in the delimited geographical area.

c. There is a control system from production to marketing with respect to quality and quantity, including physical-chemical and organoleptic control by homogeneous batches of limited volume.

d. It is prohibited to coexist in the same cellar, oil mill, or drying room with wines, oils or hams without DOCa recognition, except for products for qualified payments located in the territory.

e. It must have a cartographic delimitation, by municipalities, of the lands suitable for producing wine, oil or ham with DOCa recognition [69].

In Europe, especially in Spain, the wine industry has been a pioneer in the use of geographical names to identify food products by using the PDO label, as seen in Table 1. There are 200 PDOs, of which 96 correspond to designations of origin of wine, which represent more than 48% of the PDOs that exist in Spain and are the areas the most visited by tourists; however, Andalusia, which is the first olive oil- producing region worldwide and produces 85% of Spain's oil and 32% of the world's oil, is dominated by oil PDOs (13, representing 43%), with only 8 for wine (26.6%). From the tourism point of view, the number of tourists that visited the oil PDOs at a national level did not reach 200,000, while in 2019, Spanish enotourism generated more than 3 million tourists, with one of the most visited PDOs being Marco de Jerez located in Andalusia, which had more than a half-million visitors (Fig 1). In Andalusia, there are many products, such as oil and ham, that, from a tourist point of view, are not being sufficiently exploited but have great gastronomic potential and could be used for tourist purposes, especially in indoor areas, to create wealth.

**Table 1. Distribution of agri-food products, wines and spirits with PDOs and PGIs in Spain and Andalusia (September 2020).**

| AGRI-FOOD PRODUCTS | PDO | | PGI | |
|---|---|---|---|---|
| | Spain | Andalusia | Spain | Andalusia |
| Fresh meat (and offals) | - | - | 22 | 0 |
| Meat products | 5 | 1 | 11 | 2 |
| Cheese | 27 | - | 2 | - |
| Other animal products (honey) | 3 | 1 | 4 | - |
| Oils and fats (31 oils and 2 butters) | 31 | 13 | 3 | 1 |
| Fruits, vegetables and fresh and transformed cereals | 27 | 4 | 36 | 2 |
| Fish, seafood and fresh crustaceans and derived products | 1 | - | 4 | 4 |
| Other products (saffron, paprika, tiger nut, hazelnut, vinegar and cider) | 9 | 3 | - | - |
| Bakery, confectionary, pastry and dessert products | - | - | 16 | 4 |
| Cochineal | 1 | 0 | 0 | 0 |
| **Total PDO and PGI of agri-food products** | **104** | **22** | **98** | **13** |
| Wines with Designation of Origin (DO) | 73 | 6 | - | - |
| Wines with Qualified Design of Origin (DO Ca) | 2 | - | - | - |
| Quality Wines with Geographical Indication (QW) | 10 | 2 | - | - |
| Paid wines (PV) | 11 | - | - | - |
| Wines with Geographical Indication (GI) | - | - | 42 | 16 |
| Aromatized wines | | | 1 | 1 |
| **Total PDO and PGI of WINES** | **96** | **8** | **43** | **17** |
| Spirits with PGI | 0 | 0 | 19 | 1 |
| **Total PDO and PGI** | **200** | **30** | **160** | **31** |

Source: Prepared by the authors based on information from the Ministry of Agriculture, Food and Environment and the European Commission, Directorate General of Agriculture and Rural Development.

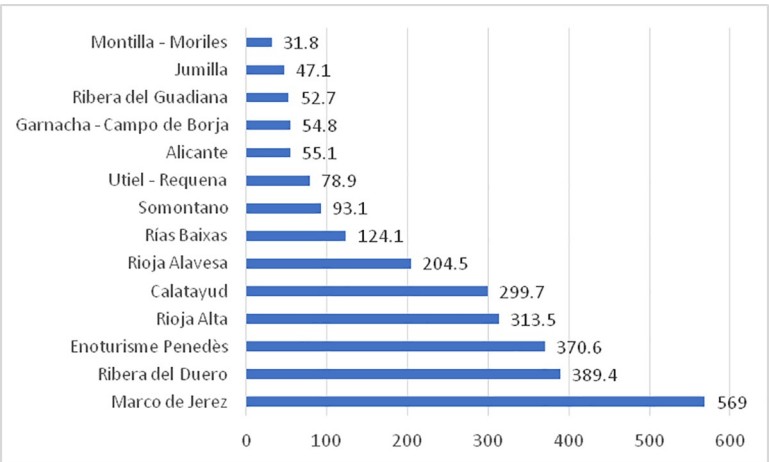

**Fig 1. Visitors of the main denominations of origin of wine in Spain (year 2019, thousands of people).** Source: Prepared by the authors based on information from the Statistics National Institute (INE).

Therefore, PDOs are key factors for tourism development because tourists increasingly demand higher quality products and seek healthy diets and activities that encourage environmental sustainability. PDOs are responsible, at the very least, for ensuring the quality of raw materials in the gastronomy sector because specific rules guide these companies. These rules are implemented by the Regulatory Council, relative to the production area, product varieties, collection practices, product preparation, bottling and quality control, and offer the perception of a certain degree of homogeneity, at least in minimum quality standards, of the brands protected under the same PDO. This quality certificate raises a series of connotations, such as recognition, quality, reputation and loyalty, that generate an intrinsic value to the PDO, which is valued by both producers and consumers. From the perspective of the producer, this value can provide economic and financial results; from the perspective of the consumer, it provides utility and can constitute in itself a decisive variable in choice, along with other variables, such as price, commercial decisions by producers and distributors [70].

Therefore, it is not by chance that the most famous food routes are those that travel the product circuits linked to the PDOs; this is due to the Lisbon Agreement [71] of 1958, which is an international agreement that sought the mutual recognition of quality names [72].

The creation of gastronomic routes aims to solve the challenges of marketing regional food products, as it is an instrument for regional food promotion. In this manner, the use of geographical quality indicators makes it easier for consumers to recognize the superiority and differentiating qualities of each product. The enhancement of the attribute or origin of a product has thus become an important marketing instrument for the commercialization of products and brands, especially if these brands belong to the agri-food sector [73]. The place of origin of a company or the origin of a product can become an important source of competitive advantage for companies that is capable of influencing consumers when valuing products or brands [74]. These hallmarks link the quality of a food to its geographical origin, where the main material is the part of the dishes that constitutes the rich "Mediterranean cuisine", which is highly appreciated by gastronomy experts, and the preparation of these dishes, the balance of flavours and raw materials constitute the gastronomic itineraries developed to experience them, known as gastronomic routes, which can be instruments used to position products and associate them with a designation of geographical quality [75]. However, many entities contribute to the design a gastronomic route, from local agents, such as producers, merchants,

restaurants and hoteliers, to public bodies, such as Provincial Councils or City Councils; these agents and bodies link tourism with food and should never ignore the annexes that link food and beverage clusters with tourism, as doing so usually leads to the loss of development and marketing opportunities for both, causing the failure of some gastronomic routes.

Among the elements that characterize a gastronomic route are (a) production that distinguishes one region from another region, (b) the itinerary developed on a road network, (c) the establishments along the route that produce, distribute or advertise the food that gives rise to the route, (d) a minimum number of members along each route to justify its opening, (e) a regulatory standard that regulates the operation of members, (f) a regional menu whose dishes have been prepared with the products that characterize the route, (g) a local organization, association or tourism office that offers information about the gastronomic route, (h) route signs and maps that provide information about the route and (i) the supply of the culinary product in restaurants and establishments in the area (Fig 2).

The development together with the success of a gastronomic route starts, first, from the following three factors: the offering of the product in the restaurants, a minimum number of members on the route and, finally, an agreement regarding the rules approved by the members. The members on the route establish the region that makes up the route and its limits, the itinerary to follow and the levels below the members. The regulation of the gastronomic route must include the complete information to be provided and its commercialization, the establishment of an association whose main objective is the maintenance and promotion of the route and, finally, the definition of the official dishes of this route.

The routes can be organized on various bases [76]:

- Gastronomic routes by product: These are routes organized on the basis of a certain product, e.g., wine, cheese, oil, etc. This is the most frequently established route type.

- Gastronomic routes by dish: These are routes organized on the basis of a prepared dish. That is, the kitchen is the common thread along the route.

- Ethnic-gastronomic routes: Although they could be integrated into the routes by dish, the ethnic component is so important that the product deserves to be distinguished. These are ventures are supported by the culinary traditions of immigrant peoples.

There can be countless activities related to the products with which a route is identified. The following may stand out:

- Visits to the producers, who provide tours of their establishments that show tourists the production processes and allow them to taste their products;

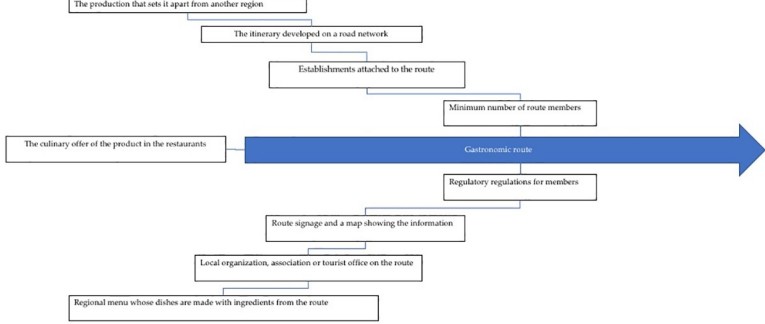

**Fig 2. Elements of a gastronomic route.** Source: Prepared by the authors based on data from the Spanish Association of Wine Cities.

- Restaurants that offer traditional dishes with local products;

- Related museums: these establishments provide information about the product and place and the history and traditions related to preparing dishes or harvesting and preparing the raw material (e.g., wine, ham and oil museums);

- Changes in consumer habits have led to a growing interest in products of higher quality that are differentiated and adapted to the new needs of different groups and market segments. Given this increase in the consumption of differentiated products based on their quality, one of the most appreciated manners of achieving this differentiation in the agri-food sector is agri-food quality certificates, which integrate in their definition not only the geographical origin but also, in a relevant manner, the tradition and specialization in regard to producing high-quality products with different characteristics [77]. Agri-food companies increase their competitiveness and achieve their market share by establishing differentiation strategies for their products, which are based on highlighting the differences in attributes, materials or characteristics with respect to a competitive product;

- Thus, interest in food products and tourism has materialized, on the one hand, with the growth of catering linked to popular cuisine and indigenous quality products and, on the other hand, in the consolidation of a new submodality of tourism, i.e., gastronomic tourism [7].

In Spain, the most commercialized routes revolve around the designations of origin of wine (Fig 3); in 2020, there were 29 Spain Product Club wine routes, with three of those routes in Andalusia.

Andalusia received more than 32 million tourists in 2019 [78], of which 12.4 million were foreign tourists. Although Andalusia is a region where sun and beach tourism predominates, gastronomy can be a differentiating element that attracts tourists and thus provides new opportunities outside of the summer season. However, from the business point of view, to generate quality tourism, it is necessary to know the profile, motivations and demands of gastronomic tourists regarding gastronomic products and understand what distinguishes the iconic products in areas or regions.

As has been observed, there have been many studies on gastronomic tourism from different perspectives, but there have been very few studies that use econometric models to estimate

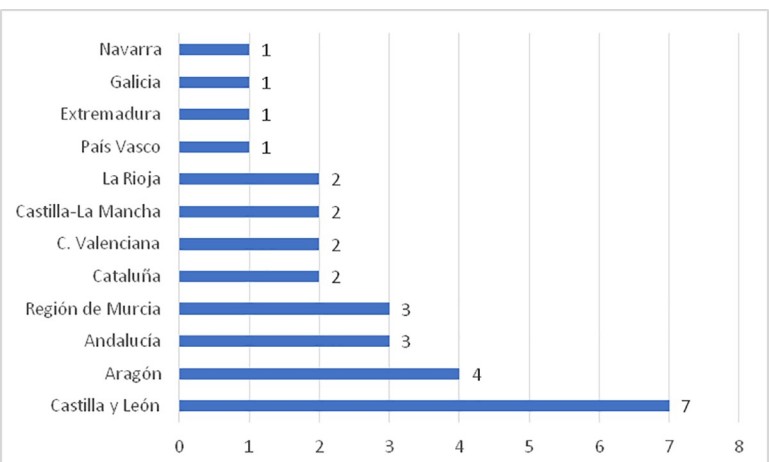

**Fig 3. Wine routes by regions (2020).** Source: Prepared by the authors based on data from the Spanish Association of Wine Cities.

demand. In this study, we will analyse the gastronomic routes in southern Spain (Andalusia) by product, with the objective of predicting the potential demands of the tourists who visit them, to establish marketing campaigns focused on less visited routes with the objectives of promoting those routes, attracting quality sustainable tourism and making gastronomy a source of income in mainly inland cities and towns.

## Materials and methods

To understand the profile of gastronomic tourists in Andalusia and the potential demand (Table 2), the following two types of information sources have been used:

- A field study aimed at the population of tourist consumers who followed a gastronomic route or visited a Protected Designation of Origin of Andalusia (Southern Spain) was conducted. A sample of 630 gastronomic tourists were recruited from February to December 2019, with the objective of determining the tourist profile. A questionnaire consisting of 23 questions was divided into the following four blocks:

    1. The first block of the questionnaire collected personal information (age, gender, educational level, marital status, etc.).

    2. The second block gathered information about the route taken (How did you learn about the gastronomic route? Did the route meet your expectations? What would you change? Did you travel expressly because of the gastronomic route? etc.).

    3. The third block addressed the motivation for gastronomic tourism (Why did you choose a gastronomic route?).

    4. The fourth block collected information regarding value (services received while following the route, price of the trip, hospitality and treatment received, etc.).
       The access by the surveyors to the gastronomic route/PDO/PGI and the conduct of interviews with tourists was authorized by the managing body and owner of the DOP´s / PGI´s. Prior to the completion of the questionnaire, tourists were informed of academic purposes and anonymity in answering. Consent to take the questionnaire was verbal. At all times, the visitor's anonymity to the gastronomic route/PDO/PGI was guaranteed.

- Information on the number of monthly tourists who visit PDOs in Andalusia (from 2015–2019), as well as the time series for the number of gastronomic tourists (National Institute of Statistics (INE)) for the same period.

**Table 2. Fact sheet for the survey.**

|  | Demand survey |
| --- | --- |
| Population | Tourists of any gender over 18 years of age who undertook/visited a gastronomic route/PDO/PGI |
| Sample size | 630 |
| Margin of error | ±4.2% |
| Confidence level | 95%; p = q = 0.5 |
| Sampling System | Simple random |
| Date of fieldwork | February 2019 –December 2019 |

Source: Own elaboration.

With the qualitative and quantitative information extracted from the questionnaire, a univariate descriptive analysis was first performed (gender, age, income level, etc.). Then, a bivariate analysis was conducted using contingency tables to analyse associations or independence between two variables ($\chi^2$ statistic), where hypothesis $H_0$ is that the analysed variables are independent and hypothesis $H_1$ is that the variables analysed are related. A third technique, a SARIMA model (used for the prediction of tourists [79–85]), was used to predict the potential demand by gastronomic tourists in Andalusia, based on a sample (61 data points) collected from January 2015 to January 2020, based on the Box–Jenkins (BJ) methodology, using ARIMA models. According to Gujarati [79], the facilitating factor of this prediction method is in an analysis of the probabilistic, or stochastic, properties of the economic time series (in this case, the number of gastronomic tourists in Andalusia). In the time series models (BJ), the gastronomic tourist variable can be explained over time by its past or lag values and by the stochastic error terms, giving ARIMA models an advantage of being less costly in data collection, as only historical observations of the data are required. In contrast, the main limitation of using univariate analysis is that it does not recognize any causal relationship with the behaviour of other endogenous variables or information related to the behaviour of other explanatory variables.

Time series models have been widely used in forecasting tourism demand with a predominance of ARIMA models [80]. According to published studies by Song and Li [81], the different versions of the ARIMA models proposed by Box and Jenkins [82] to identify, estimate and diagnose dynamic models of time series have been applied in most post-2000 studies that used time series forecasting techniques. In the case of seasonal time series analysis, these models are called SARIMA, and they are differentiated from stationary ARIMA models in that the latter consider the mean of the series to be constant over time, and the correlation function depends on the lag and not on the time in which it is calculated. However, the time series, in addition to random, cyclical and seasonal variations, present a trend and seasonal components (the mean varies over time and by seasons), which makes the stationary processes unsuitable for modelling. For this reason, integrated models are introduced, thus eliminating the trend and seasonal component of these models.

The SARIMA models (p,d,q) × (P,D,Q)s are described by the following expressions:

$$\varphi\,(B)\,\Phi(B^s)\,Z_t = \theta\,(B)\,\Theta(B^s)\,a_t$$

$$Z_t = (1 - B)^d\,(1 - B^s)^D\,Y_t^{(\lambda)}$$

Where the operators introduced in the formulas are $Y_t$ (series observed, in our case, it is the gastronomic tourism demand, $\lambda$ (represents the correction of the trend in variance of the series), $Z_t$ (series that is de-seasonalized and without a trend, that is, is stationary), B (lag operator), $(1 - B)$ (typical difference operator), $B^s$ (seasonal lag operator, $(1 - B^s)$: seasonal difference operator). The difference operators and seasonal difference operators, in general, eliminate the trends and the seasonal components of the series, respectively. $\phi(B)$ is the autoregressive polynomial of order p, corresponding to the ordinary part of the series; $\theta(B)$ is the polynomial of moving averages of order q, corresponding to the ordinary part of the series; $\Phi(B^s)$ is the p-order autoregressive polynomial, corresponding to the seasonal part of the series; $\Theta(B^s)$ is the polynomial of moving averages of order Q, corresponding to the seasonal part of the series; at is the disturbance of the model; and D is the number of times the seasonal difference operators and typical difference are applied to the original series to make it stationary.

In the ARIMA and SARIMA models, the behaviour of a time series is explained from the past observations of the series itself and from the past forecasting errors. Several studies have

shown how ARIMA models and their different variants obtain good results in forecasting tourism demand and, in most cases, surpass other time series methods, such as the periodic autoregression model, moving averages, smoothed exponential, non-causal basic structural model and multivariate adaptive regression splines [83–85], based on these authors who support the use of ARIMA models for predicting tourism demand, has served as an argument in this study to use the models in predicting gastronomic tourism demand.

The tourist segment dedicated to gastronomy is not sufficiently exploited and because gastronomic tourism needs to be stimulated, studies are needed to indicate its evolution, as well as to apply the necessary marketing tools to deseasonalize the sun and beach tourism characteristic of Andalusia.

## Results

Based on the sample surveyed (Table 3), the predominant profile of gastronomic tourists in Andalusia was as follows: male (54.6%), age 50 to 59 years (38.6%), secondary education (64.3%), average income between 1,500 and 2,000 euros (35.1%), works for others (67.7%), married (48.1%), travels gastronomic routes with a partner (43.2%) and from Andalusia. There was a significant difference among products related to the income levels of enotourists and other gastronomic tourists (olive oil, ham and pastry tourism) (Kruskal-Wallis = 0.009). Enotourists had greater purchasing power; 32% of the enotourists had a monthly income greater than € 2,500, while only 5% of the remaining gastronomic tourists reached this income level. Overall, 12.4% of the participants had a monthly income greater than € 2,500. If we analyse similar studies of gastronomic tourist profiles, such as that by Mealhada-Portugal [86], some variables coincide, such as a higher percentage of men (56%) and education level; in a study conducted [87] in the Dominican Republic, 56% were male, but the average age of gastronomic tourists was different from that in this study. In Andalusia, tourists between 50 and 59 years of age (38.6%) with an average purchasing power predominate; in a study by Orgaz and Lopez, younger tourists, from 30 to 39 years of age, with low purchasing power predominate.

Among those who engage in gastronomic tourism in Andalusia, the predominant place of residence was the Autonomous region of Andalusia, i.e., 72.4%, indicating proximity tourism, with 53.4% not staying overnight at the destination (53.4%). The average daily expenditure of gastronomic tourists was higher than that of general tourists who visited the region [88]; the latter was € 68 per day in 2019, and the average daily expenditure of gastronomic tourists in this study was between € 65 and € 100. Among the gastronomic tourists, those who came from abroad reported an average daily expenditure of more than € 250 because they spent the night and ate. It is in this market where efforts should be made to attract more gastronomic tourists from outside Spain instead of local gastronomic tourists, whose spending is below the average expenditure of those from outside of Andalusia.

This type of tourism is mainly promoted through the internet or social networks (50.5%); however, more than 25% of the tourists learned of a route based on the recommendation of friends and family (Table 4).

Because more than 68% of people would repeat the route and the degree of satisfaction exceeded 75% for more than 80% of the respondents, the product offered has high quality, similar to the results obtained in studies regarding Valencian gastronomy [89], in which 41% of the respondents considered it very good and 56% considered it good. To increase profitability, marketing at the national and international levels is recommended. Although there are routes associated with denominations of origin, such as the wine of Jerez, which receives approximately half a million tourists a year, there are others, such as the Denomination of Origin of

**Table 3. Tourist profile of gastronomic tourism in Andalusia.**

| Block | Factor | Classification | Percentage |
|---|---|---|---|
| **Personal characteristics of gastronomic tourists** | **Age** | 18–29 years old | 15.2% |
| | | 30–39 years old | 18.4% |
| | | 40–49 years old | 20.5% |
| | | **50–59 years old** | **38.6%** |
| | | More than 60 years old | 7.3% |
| | **Education level** | No studies completed | 0.5% |
| | | Primary studies | 17.3% |
| | | **Secondary studies** | **64.3%** |
| | | Higher studies | 18.0% |
| | **Gender** | **Male** | **54.6%** |
| | | Female | 45.4% |
| | **Marital status** | Single | 28.3% |
| | | Married | 48.5% |
| | | Divorced/separated | 22.7% |
| | | Other status | 0.5% |
| | **Level of monthly income of the family unit** | Less than 1000 euros | 13.2% |
| | | 1001–1500 euros | 18.7% |
| | | **1501–2000 euros** | **35.1%** |
| | | 2001–2500 euros | 20.6% |
| | | + than 2500 euros | 12.4% |
| | **Whom did you travel with?** | Alone | 6.3% |
| | | **Accompanied by my partner** | **43.2%** |
| | | With friends | 9.0% |
| | | With family members | 41.5% |
| | **Where are you from?** | Andalusia | 72.4% |
| | | **Rest of Spain (except Andalusia)** | **23.1%** |
| | | European Union (except Spain) | 4.1% |
| | | Rest of the world (except European Union) | 0.4% |
| | **Employment status** | **Employed by others** | **67.7%** |
| | | Self-employed | 24.3% |
| | | Retired | 5.6% |
| | | Unemployed | 2.1% |
| | | Student | 0.3% |
| | **Duration of the trip** | **Less than 24 hours** | **53.4%** |
| | | 2–3 days | 35.3% |
| | | More than 3 days | 11.3% |
| | **Daily expenditure** | Less than 30 euros | 5.2% |
| | | 30–64 euros | 28.4% |
| | | **65–100 euros** | **49.3%** |
| | | More than 100 euros | 17.1% |

Source: Own elaboration.

Pedroches Ham, that only receives 15,000 tourists a year. By taking advantage of the knowledge that tourists have of some routes, it may be possible to combine routes for various products to further promote tourism.

In terms of the expectations that tourists had regarding the route, 94.3% reported that the route met their expectations, indicating high value for the product offered. However, the

**Table 4. Univariate analysis results for the survey of gastronomic tourists in Andalusia: Questions about the visit.**

| Block | Question | Classification | Percentage |
|---|---|---|---|
| Questions about the visit | How many people are with you on this trip? | 1 person | 4.1% |
| | | **2 to 4 people** | **73.4%** |
| | | More than 4 people | 22.6% |
| | Has the PDO or gastronomic route met your expectations? | **Yes** | **94.3%** |
| | | No | 5.7% |
| | What would you improve? | Nothing | 10.1% |
| | | **Signage** | **37.2%** |
| | | Explanation of the route or the PDO | 26.4% |
| | | More audio-visual media | 15.1% |
| | | Other | 11.2% |
| | Would you be interested in receiving more information after the visit? | **Yes, if it is free of charge.** | **52.1%** |
| | | Yes, in any case | 28.3% |
| | | I do not consider it necessary | 19.6% |
| | Did you come expressly because of this gastronomic route or did you learn of it while in Andalusia? | **I came expressly because of the route** | **58.1%** |
| | | I learned about it in Andalusia | 41.7% |
| | Does the price paid seem reasonable? | **Yes** | **90.2%** |
| | | No | 9.8% |
| | How did you learn about the route? | Travel agency | 18.4% |
| | | **On the internet, through social networks** | **50.5%** |
| | | Recommended by friends and family | 25.3% |
| | | Brochures | 2.6% |
| | | Other media | 3.2% |
| | Would you try a similar route in the future? | **Yes** | **68.1%** |
| | | No | 31.9% |
| | Were you satisfied with the visit? | less than 25% | 1.6% |
| | | 25% - 50% | 2.1% |
| | | 51–75% | 4.7% |
| | | 76% - 99% | 36.2% |
| | | **100%** | **55.4%** |

Source: Own elaboration.

experience could be improved by adding more signs; some tourists found it very difficult to locate some wineries or oil mills and got lost along the way. In addition, more audio-visual media explaining the production of the product (15.2%) or brochures that explain the destination (26.42%) may make the trip more enjoyable for tourists.

Notably, 58.3% of the tourists travelled specifically because of the gastronomic route, making it the main objective of the trip, and 90.2% reported that the price was reasonable for the quality of the tourist product.

The primary motivation for gastronomic tourism (Table 5) is to learn about the production of wine, oil, ham, etc. (58.6%), with wineries, oil mills or drying facilities being the main tourist attractions [29, 35, 63], where at the end of the visit, the tourist was able to taste the product and appreciate the quality of the gastronomic product. Furthermore, 98.1% of the respondents would be satisfied with the creation of a combined route for gastronomic products because they think that doing so will enrich the gastronomy, making it more attractive through the pairing of products, e.g., wine-ham, oil-cheese, etc.

**Table 5. Univariate results for the survey of gastronomic tourists in Andalusia: Motivation.**

| Questions about the motivation for the visit | What motivated you to visit? | Learn the culinary tradition of the place | 32.5% |
|---|---|---|---|
| | | Learn the process of making wine, oil, ham, etc. | 58.6% |
| | | Attend food festivals | 6.2% |
| | | Visit ham, oil and wine museums. | 2.7% |
| | How do you assess the tourism management at the sites you have visited? | **Good** | **64.1%** |
| | | Regular | 30.3% |
| | | Bad | 5.6% |
| | What do you think about the creation of a combined route for various gastronomic products and theatrical productions at destinations? | **I agree** | **98.1%** |
| | | I do not agree; I prefer single gastronomic routes, not combined | 1.9% |

Source: Own elaboration.

Regarding the assessment of tourist management at the places visited, 60.4% reported that the management was good, and 98.1% indicated that routes with multiple gastronomic products should be offered, as well as theatrical representations of the period represented; in some inland towns of the province of Cordova where there are Roman sites, theatrical productions are part of the route, with very high acceptance among gastronomic tourists [66].

Based on the univariate analysis results, we can highlight that the tourists in Córdoba and the province were very satisfied with the product, were knowledgeable about the subject and believed that combined products should be created. There was also appreciation regarding the lack of coordination in the management of the product in part by having visited similar sites. Therefore, Córdoba has great market potential for this tourism niche, but it is necessary to adequately manage and coordinate public and private entities to offer a quality product and increase tourism deals, especially at night, which will increase overnight stays and the average expenditure of tourists; these are two handicaps in Córdoba if the actions that are carried out are individual and not coordinated.

If we analyse the valuation provided by gastronomic tourists in Andalusia (Fig 4) and compare that with the valuation provided by tourists in general (sun and beach, cultural tourism,

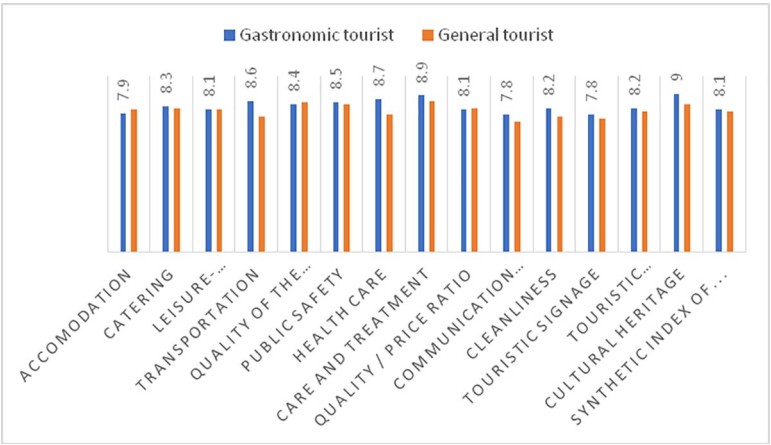

**Fig 4. Comparison of the valuations by gastronomic tourists in Andalusia with respect to table 2019.** Source: Prepared by the authors based on data from the Ministry of Tourism and Sports of Junta de Andalucía.

etc.), on average, gastronomic tourists had a better perception index for the tourist product (8.1 compared to 8), with very positive assessments of the treatment received (8.9) and the catering (8.3). Andalusia also obtained high value ratings for health care. In the current situation, after having passed the first wave of COVID-19, feeling protected at a tourist destination is a positive incentive to travel. Gastronomic tourists place higher value on Andalusia than do general tourists.

With the objective of deepening the analysis among the different variables, bivariate analysis was performed. There was a strong relationship between age and the degree of satisfaction ($\chi^2$ = 235.88, p = 0.003), as older tourists placed higher value on the route. Age also influenced the knowledge regarding the route ($\chi^2$ = 34.32, p = 0.0006); younger tourists used new technologies, the internet, social networks and tourism websites, while older tourists received recommendations from friends and family. There was also a strong relationship between the reason for taking the route and age ($\chi^2$ = 64.5, p = 0.00).

However, there was no relationship between visit motivation and gender ($\chi^2$ = 0.11, p = 0.99) or income level ($\chi^2$ = 5.47, p = 0.94).

Additionally, the education level and degree of satisfaction were related; tourists with higher education levels had lower degrees of satisfaction because either the explanations regarding the processing of products were not correct (lack of professionalism) or they believed that more audio-visual media were needed ($\chi^2$ = 24.05, p = 0.02).

Gastronomic tourism is a niche market that is not sufficiently exploited in Andalusia and therefore has great potential for development, which, combined with other forms of cultural tourism, could generate wealth and expand the cultural offerings of Andalusia [12, 19, 52].

## ARIMA for predicting the demand for gastronomic tourism in Andalusia

Andalusia is the leading region in Spain in terms of sales of agri-food products abroad, which have a positive impact on the economy of the region and on the number of gastronomic tourists who travel to this region to learn about production processes. Despite the existence of this tourist segment, gastronomic tourism is not being sufficiently exploited, mainly among international tourists, because there are very few tourists compared with those in other already established international gastronomic destinations, such as France or Italy. The Basque Country and Catalonia are the most popular gastronomic destinations in Spain, with Andalusia being less prominent. To try to improve this situation, Junta de Andalucía has launched an advertising campaign named "Tasty Andalucía", which is aimed at international tourists who visit the region to promote the agri-food and fishery products of the region, especially products covered by quality labels, such as PDOs, PGIs and organic products. With the aim of promoting gastrotourism, advertising has been conducted through the internet, social networks and other actions targeting gastronomy audiences at the international level so that the products are also consumed by the tourists when they return to their respective countries of origin.

However, what is the potential demand for gastronomic tourism in Andalusia? Can it be deseasonalized? These are necessary questions that need to be answered. For this, studies that evaluate the evolution of gastronomic tourism in Andalusia are essential, as well as the application of the necessary marketing tools to stimulate such tourism.

There are no studies that forecast the potential demand for gastronomic tourism in Andalusia; however, there are studies on products such as ham [90] oil, [54, 91] and wine [28, 48]. This work is a novel contribution that fills this research gap, which mainly exists due to the difficulty of obtaining data on the number of gastronomic tourists, as the statistics available to companies that are dedicated to tourism are scarce. In this research, information regarding tourists who travel different gastronomic tourism routes (especially those for wineries, oil

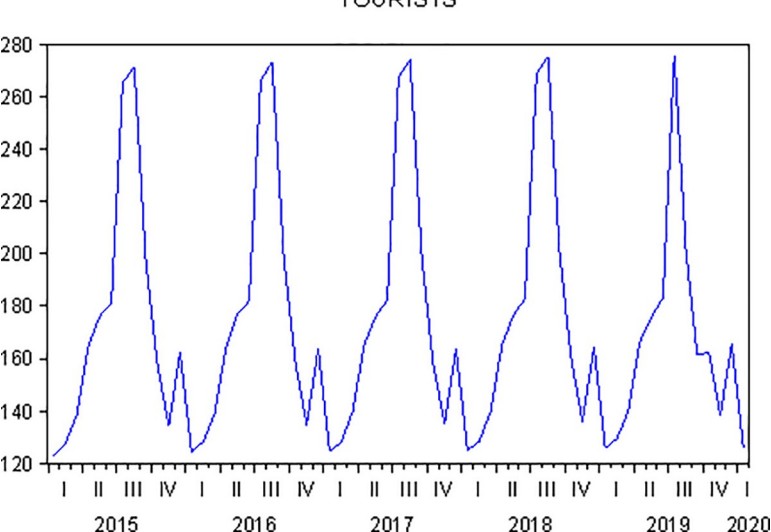

**Fig 5. Evolution of the demand for gastronomic tourism in Andalusia (January 2015—January 2020).** Source: Own elaboration.

mills, dryers and restaurants) was collected. ARIMA models were used to predict the tourist demand [83, 84, 91].

Fig 5 shows a slightly growing trend for gastronomic tourism demand in the 5 years analysed (January 2015 to January 2020). There was also variance, which was corrected with Box-Cox transformation, $\lambda = 0.3$; additionally, the mean trend and cycle trend were corrected by calculating the difference between the mean and cycle trends.

The SARIMA model [(1,1,10) (0,1,0) 12] for the estimated monthly demand for gastronomic tourism was as follows (Table 6):

$(1-0.364731\ B)\ (1\text{-}B)^1\ (1\text{-}B^{12})^1\ \text{Tourists}^{0.3} = (1+1.376113)\ at$

$t_{\phi 1} = 12.09682^*\ t_{\theta 1} = -6.918053^*$

$^*$ Significant parameters for $\alpha = 0.05$.

Tables 7 and 8 show the different validation tests, such as the Ljung-Box test, for the model. The null hypothesis of the absence of an autocorrelation is met; i.e., the probability is higher than the 1% significance level (Prob. column).

The ARCH statistic (Table 9) indicates that in the model, there is no autoregressive conditional heteroscedasticity (null hypothesis); i.e., the probability (0.2045) is greater than the 5% significance level.

**Table 6. Bivariate analysis.**

| Associated variables | $\chi^2$ | df | P-value |
|---|---|---|---|
| Age of the tourist/degree of satisfaction with the route taken | 35.88 | 16 | 0.003 |
| Education level/degree of satisfaction with the route conducted | 57.6 | 12 | 0.0 |
| Age of the tourist/knowledge of the route | 34.32 | 12 | 0.0 |
| Age of the tourist/reason for taking the route | 64.5 | 9 | 0.00004 |

$\chi^2$, chi-square statistic; related variables, $\alpha = 0.05$; df = degrees of freedom. Source: Own elaboration.

**Table 7. Estimation of the demand for gastronomic tourism in Andalusia.**

| Dependent Variable: D(TURISTAS^0.3,1,12) | | | | |
|---|---|---|---|---|
| Method: least squares | | | | |
| Sample (adjusted): 2016 M03 2020 M01 | | | | |
| Variable | Coefficient | Std. Error | t-Statistic | Prob. |
| AR(1) | 0.364731 | 0.030151 | 12.09682 | 0.0000 |
| MA(1) | -1.376113 | 0.198916 | -6.918053 | 0.0000 |

Source: Own elaboration.

In Fig 6, the predictions obtained for the year 2021 and their comparison with the year 2019 are provided. The year 2020 was omitted because it is an atypical year; i.e., due to the pandemic, the closing of borders and restriction of movement within Spain, it was not possible to travel in the months of March to June 2020. These predictions are made under the assumption that the pandemic is controlled and that the borders are fully open, with no restrictions on tourist entry.

It is expected that during the year 2021, there will be slight growth in gastronomic tourism, especially for the national market because many people are still hesitant regarding travelling abroad due to the fear of closing borders in the face of a possible new wave of the pandemic;

**Table 8. Ljung-Box statistics.**

| Sample: 2015 M01 2020 M01 | | | | | | | |
|---|---|---|---|---|---|---|---|
| Included observations: 47 | | | | | | | |
| Q-statistic probabilities adjusted for 2 ARMA terms | | | | | | | |
| Autocorrelation | Partial Correlation | | AC | CAP | Q-Stat | Prob | |
| . \|. \| | . \|. \| | 1 | 0.016 | 0.016 | 0.0126 | | |
| *** \|. \| | *** \|. \| | 2 | -0.364 | -0.365 | 6.8020 | | |
| . \|. \| | . \|. \| | 3 | -0.041 | -0.031 | 6.8884 | 0.009 | |
| . \|. \| | ** \|. \| | 4 | -0.057 | -0.218 | 7.0610 | 0.029 | |
| . \|. \| | . * \|. \| | 5 | -0.038 | -0.077 | 7.1423 | 0.067 | |
| . \|. \| | . * \|. \| | 6 | -0.003 | -0.134 | 7.1429 | 0.129 | |
| . \|. \| | . * \|. \| | 7 | -0.011 | -0.089 | 7.1503 | 0.210 | |
| . \|. \| | . * \|. \| | 8 | 0.007 | -0.084 | 7.1534 | 0.307 | |
| . \|. \| | . * \|. \| | 9 | 0.008 | -0.070 | 7.1577 | 0.413 | |
| . \|. \| | . * \|. \| | 10 | -0.024 | -0.094 | 7.1941 | 0.516 | |
| . \|. \| | . \|. \| | 11 | -0.003 | -0.065 | 7.1947 | 0.617 | |
| . \|. \| | . * \|. \| | 12 | 0.007 | -0.074 | 7.1978 | 0.707 | |
| . \|. \| | . * \|. \| | 13 | -0.005 | -0.067 | 7.1996 | 0.783 | |
| . \|. \| | . * \|. \| | 14 | -0.002 | -0.068 | 7.1999 | 0.844 | |
| . \|. \| | . \|. \| | 15 | 0.003 | -0.061 | 7.2004 | 0.892 | |
| . \|. \| | . \|. \| | 16 | -0.002 | -0.065 | 7.2008 | 0.927 | |
| . \|. \| | . \|. \| | 17 | 0.007 | -0.050 | 7.2046 | 0.952 | |
| . \|. \| | . \|. \| | 18 | 0.006 | -0.052 | 7.2075 | 0.969 | |
| . \|. \| | . \|. \| | 19 | -0.006 | -0.053 | 7.2101 | 0.981 | |
| . \|. \| | . \|. \| | 20 | -0.002 | -0.051 | 7.2106 | 0.988 | |

Source: Own elaboration.

**Table 9. ARCH test for heteroscedasticity.**

| Heteroskedasticity Test: ARCH | | | |
|---|---|---|---|
| F-statistic | 2.758233 | Prob. F(1,44) | 0.1039 |
| Obs * R-squared | 2.713506 | Prob. chi-square (1) | 0.0995 |

Source: own elaboration.

therefore, the national market will improve the data for the sector. As can be observed, July and August are the months with the greatest number of gastronomic tourists, coinciding with the summer holidays; there is also an increase in tourists in December, coinciding with the Christmas holidays. Although gastronomic tourists generally take day trips and do not stay overnight, complementary activities, especially at night, should be promoted. Thus, destinations can provide attractive offerings that allow tourists to spend the night, thus, increasing the average spending and eliminating the strong seasonal tourism in Andalusia. Historically, sun and beach tourism constitutes 80% of the sector, with coastal areas being underutilized from October to April. Gastronomy and cultural heritage, both tangible and intangible, can mitigate this trend and attract tourists throughout the year.

## Discussion

Gastronomy is increasingly motivating tourists' travel intentions, with gastronomic destinations becoming increasingly visited places, decreasing the seasonality of sun and beach tourism. This statement coincides with studies from [88–90, 92, 93].

There is, however, no regional brand that characterizes gastronomy as outlined in the research [16], which in turn could attract a larger volume of tourists. Nevertheless, a collaboration between public organizations and private entities, similar to that existing in Nordic countries [94], aims to actively promote the Andalusian gastronomic heritage. Under this gastronomic basis and since 2018, the Junta de Andalucía, local authorities and private businesses, have elaborated marketing campaigns such as "Andalucía, Paisajes con Sabor", aiming to increase the volume of tourists that visit the region. This target consumer group of tourists currently exceeds 3 million people, and primarily come to enjoy cuisine, oenological wealth,

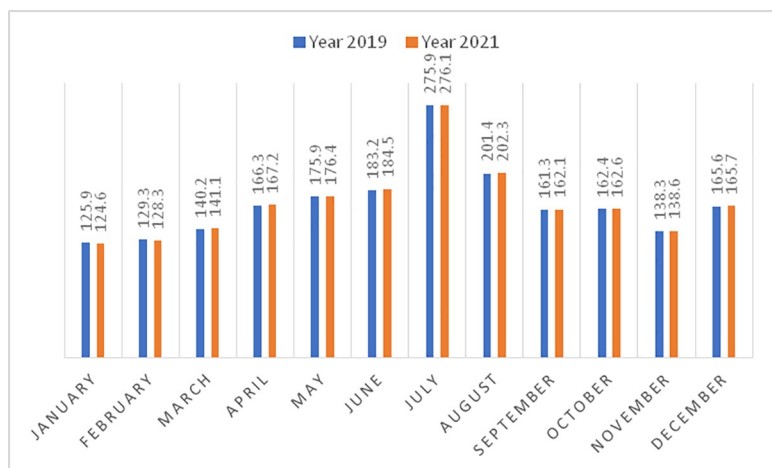

**Fig 6. Prediction of the demand for gastronomic tourism in Andalusia (2021 and its comparison with the year 2019).** Source: own elaboration.

olive grove and the landscape of inland Andalusia, showing promising potential to propel the economic development in the region as has happened in other Spanish regions [95].

Although there are gastronomic routes created around these products protected by quality labels, these routes are not well visited because these products do not have defined gastronomic routes, such as those for extra virgin olive oil, traditional dishes of Jewish or Arab origin and singular raw materials [91], as shown by the results obtained, which are similar to the studies carried out in the region of Extremadura [96] or Catalonia (Spain) [54].

This study shows that the gastronomic tourists studied have very similar motivations to those who visit other gastronomic destinations, which mainly include learning about the manufacturing process and tasting the product, especially wine [2, 14, 20]; additionally, they have a higher average income, and their average expenditure per day is higher than that of sun and beach tourists, as they are mature tourists (50 to 59 years of age); however, in other gastronomic destinations, the tourists are younger [82] because the destination, in addition to gastronomy, is complemented by aquatic and sports activities.

Investing in the development of local activities in Andalusia that combine traditional food and tourism could yield long-term, sustained gains for rural areas in the region, as shown in a similar case study in Poland [64]. This is particularly evident considering Andalusian cuisine combines products (ham, oil, wine, etc.) in sophisticated dishes to provide a unique gastronomic experience, which are increasingly in high demand by both national and foreign tourists, as is the case in other countries [55–57].

## Conclusions

This study highlights that there are gastronomic products, such as olive oil and ham, in Andalusia (Spain) that are part of the rich Mediterranean cuisine and are protected by PDOs; these resources are not being valued from the tourist point of view, causing a loss of economic opportunity, mainly in rural inland areas [1].

It is important to understand the motivations, profiles and demands of gastronomic tourists and to create a tourist products according to their needs. However, it is necessary to increase gastronomic tourism in Andalusia because it is currently low. Some businesses, such as hotels or restaurants, are not profitable because the predominant type of tourism in Andalusia is that of locals, and thus, they do not spend the night and spend little on eating, although they do buy food products when they visit wineries, oil mills or dryers, purchasing souvenirs, wine, oil or ham. It is necessary to promote Andalusian gastronomy in markets other than Spain, as occurs in other Spanish communities such as Extremadura, where gastronomic tourists tend to spend the night [93], increasing the average daily expenditure. To compete with an attractive tourism product in both national and international markets, a consistent and solid offer adapted to different motivations is needed, requiring private and public coordination and concrete actions to increase and enhance the gastronomic routes in Andalusia. It is necessary to attract investors or "business angels" who are willing to create wealth and gamble on gastronomy through tourism as a business opportunity. In addition, Andalusian gastronomy should be offered on social networks, since the majority of gastronomic tourists to Andalusia learn about the gastronomic routes through the internet, and according to Rodriguez et al [94], Facebook is a vehicle that has positioned gastronomic products that attract tourists in other countries. In the case of Andalusia, it is essential to depend on the national tourist market in times of crisis because in the previous economic crisis for Spain and especially Andalusia, the economic engine that improved the balance of payments was foreign tourism.

The predictions obtained with the SARIMA model indicate a slight increase in demand, with this increase in tourists being an opportunity for the economies of rural areas, which

represent the majority of the municipalities in Andalusia. After the COVID-19 pandemic, tourists may prefer less crowded places, located mainly in areas in contact with nature [97–99] and where they have more personalized attention and contribute to environmental sustainability; gastronomy in Andalusian is a good example of this.

It can be concluded that the culinary wealth of the Andalusian region constitutes a heritage that has not yet been sufficiently exploited. The efforts of many public bodies (Junta de Andalucía and Ayuntamientos) and private entities (designations of origin, wineries, cooperatives, hotels, restaurants, gastronomic associations, etc.) have not translated into a considerable increase in the number of gastronomic tourists, although those who participate are very satisfied and anticipate repeating their experience. The motivations for gastronomic tourism vary and depend in part on the profile of tourists, but what has been shown is that currently in Andalusia, gastronomic tourists are mainly from the region, and there are very few international tourists. Many tourists are concentrated in coastal areas, were Mediterranean gastronomy is appreciated, and visit wineries and oil mills. Currently, wine and wineries generate more than 25% of gastronomic tourism.

Gastronomy is a complement to other motivations for tourism, serving as a way to improve life and as a reminder of lived experiences. This tourist segment, considered strategic for both territorial and economic development, should be promoted and consolidated, given its high added value. In short, olive oil tourism, wine tourism and ham tourism are products with great growth potential that can favour the economic and territorial development of some Andalusian inland regions and constitute quality tourism offerings that meet the expectations of visitors.

For future lines of research, this study could be carried out in other destinations in Spain such as Canary Islands or Balearic Islands, and the results obtained in this work could be compared with those of other destinations. Another possible line of research could be to perform this same study in Andalusia, but aimed at international tourists, in order to examine their motivations and thus establish a segmentation of the touristic offerings of the community according to the type of tourist, national or international.

## Supporting information

**S1 Survey. Survey analysis of gastronomic tourism.**
(DOC)

**S1 Data. Eviews.**
(WF1)

## Author Contributions

**Conceptualization:** Mª Genoveva Millán Vázquez de la Torre.

**Data curation:** Mª Genoveva Dancausa Millán.

**Formal analysis:** Ricardo Hernández Rojas.

**Investigation:** Mª Genoveva Dancausa Millán, Mª Genoveva Millán Vázquez de la Torre, Ricardo Hernández Rojas.

**Methodology:** Mª Genoveva Dancausa Millán.

**Resources:** Mª Genoveva Dancausa Millán.

**Software:** Mª Genoveva Millán Vázquez de la Torre, Ricardo Hernández Rojas.

**Supervision:** Mª Genoveva Dancausa Millán, Mª Genoveva Millán Vázquez de la Torre.

**Validation:** Mª Genoveva Dancausa Millán, Mª Genoveva Millán Vázquez de la Torre.

**Visualization:** Ricardo Hernández Rojas.

**Writing – original draft:** Mª Genoveva Dancausa Millán.

**Writing – review & editing:** Mª Genoveva Dancausa Millán, Mª Genoveva Millán Vázquez de la Torre, Ricardo Hernández Rojas.

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
