## [Decision Letter · Decision Letter 0]

8 Jan 2021

PONE-D-20-39678

Analysis of the demand for gastronomic tourism in Andalusia (Spain)

PLOS ONE

Dear Dr. Millán Vázquez de la Torre,

Thank you for submitting your manuscript to PLOS ONE. After careful consideration, we feel that it has merit but does not fully meet PLOS ONE’s publication criteria as it currently stands. Therefore, we invite you to submit a revised version of the manuscript that addresses the points raised during the review process.

We look forward to receiving your revised manuscript.

Kind regards,

Prof. Arkadiusz Piwowar

Wroclaw University of Economics and Business

Academic Editor

PLOS ONE

Journal Requirements:

2. Please ensure that you refer to Figure 2 and 4 in your text as, if accepted, production will need this reference to link the reader to the figure.

3. We note you have included a table to which you do not refer in the text of your manuscript. Please ensure that you refer to Table 2 and 3 in your text; if accepted, production will need this reference to link the reader to the Table.

Reviewers' comments:

Reviewer's Responses to Questions

5. Review Comments to the Author

Reviewer #1: The reviewed article has been well prepared in terms of its content and methodology. The introduction, however long (maybe even too extensive ...), places the reader well in the realities of research. Well-chosen methodology (adequate to the research subject).

I wonder about the literature review - are the author / authors deliberately omitting similar research from European countries? I am thinking in particular of countries with a fragmented structure of farms, for which culinary tourism and culinary heritage are an important alternative to the development and maintenance of the character of rural areas - Italy, Romania, Lithuania, Poland. There are quite a few studies from these areas.

I would recommend supplementing this part of the article with e.g. publications such as:

Chiodo, E.; Giordano, L.; Tubi, J.; Salvatore, R. Wine Routes and Sustainable Social Organization within Local Tourist Supply: Case Studies of Two Italian Regions. Sustainability 2020, 12, 9388.

Niedbała, G.; Jęczmyk, A.; Steppa, R.; Uglis, J. Linking of Traditional Food and Tourism. The Best Pork of Wielkopolska—Culinary Tourist Trail: A Case Study. Sustainability 2020, 12, 5344.

Minta, S., 2015, Regional food products: only for tourists or also for residents. Poljoprivreda i šumarstvo. 2015. Vol. 61, no. 1, p. 51–58. DOI 10.17707/AgricultForest.61.1.06.

Białogowska, A. Culinary tourism as an important, intercultural issue. Sci. Rev. Phys. Cult. 2014, 4, 14–23.

Bessiere, J.; Tibere, L. Traditional food and tourism: French tourist experience and food heritage in rural spaces. J. Sci. Food Agric. 2013, 93, 3420–3425.

Richards, G. Food and the tourism experience: major findings and policy orientations. In Food and the Tourism Experience; Dodd, D., Ed.; OECD: Paris, 2012; pp. 13–46.

Ab Karim, S.; Chi, C.G.-Q. Culinary Tourism as a Destination Attraction: An Empirical Examination of Destinations’ Food Image. J. Hosp. Mark. Manag. 2010, 19, 531–555.

Of course, I do not impose on the author / authors the use of all three items, but it is worth getting acquainted with them, especially in the context of the solutions and recommendations sought, as well as using the experiences of other European countries.

Reviewer #2: The article presented deals with a very interesting and important topic from the economic point of view.

The article is well structured, although I recommend the following changes:

- Autonomous Community, as such, does not exist in English. I recommend using the word "region".

- I think it is convenient to separate the discussion section from the conclusions of the work, since they are totally different aspects.

- Reinforce the literature and bibliography in English.

---

## [Author Response · Author response to Decision Letter 0]

11 Jan 2021

Dear editor:

We are grateful to Plos One for giving us the opportunity to publish our article “Analysis of the demand for gastronomic tourism in Andalusia (Spain)” .

We have made the changes to the manuscript to address the issues reviewers have raised.

We have removed the figures from within the manuscript file. 

We have provide additional details regarding participants consent, in the setion methodos. 

The authors have not received specific funding for this work.

Kind regards, 

Genoveva Millán

Editor:

 Please ensure that you refer to Figure 2 and 4 in your text as, if accepted, production will need this reference to link the reader to the figure. References to figures 2 and 4 have been included in the text

. We note you have included a table to which you do not refer in the text of your manuscript. Please ensure that you refer to Table 2 and 3 in your text; References to tables 2 and 3 have been included in the text

Reviewers' comments:

We appreciate the reviewer's comments and suggestions and have made them 

Reviewer #1: The reviewed article has been well prepared in terms of its content and methodology. The introduction, however long (maybe even too extensive ...), places the reader well in the realities of research. Well-chosen methodology (adequate to the research subject).

I wonder about the literature review - are the author / authors deliberately omitting similar research from European countries? I am thinking in particular of countries with a fragmented structure of farms, for which culinary tourism and culinary heritage are an important alternative to the development and maintenance of the character of rural areas - Italy, Romania, Lithuania, Poland. There are quite a few studies from these areas. The references indicated by the reviewer have been included in the text. 

Reviewer #2: 

Dear Reviewer:

We appreciate the reviewer's comments and suggestions and have made them 

The article presented deals with a very interesting and important topic from the economic point of view.

The article is well structured, although I recommend the following changes:

- Autonomous Community, as such, does not exist in English. I recommend using the word "region". 

The word autonomous community has been changed to region.

- I think it is convenient to separate the discussion section from the conclusions of the work, since they are totally different aspects. 

The discussion has been separated from the conclusions

- Reinforce the literature and bibliography in English. New bibliography has been introduced in English

---

## [Decision Letter · Decision Letter 1]

18 Jan 2021

Analysis of the demand for gastronomic tourism in Andalusia (Spain)

PONE-D-20-39678R1

Dear Dr. Millán Vázquez de la Torre,

We’re pleased to inform you that your manuscript has been judged scientifically suitable for publication and will be formally accepted for publication once it meets all outstanding technical requirements.

Kind regards,

Prof. Arkadiusz Piwowar

Wroclaw University of Economics and Business

Academic Editor

PLOS ONE

---

## [Editor Report · Acceptance letter]

21 Jan 2021

PONE-D-20-39678R1 

Analysis of the demand for gastronomic tourism in Andalusia (Spain) 

Dear Dr. Millán Vázquez de la Torre:

I'm pleased to inform you that your manuscript has been deemed suitable for publication in PLOS ONE. Congratulations! Your manuscript is now with our production department. 

Kind regards, 

on behalf of

Professor Arkadiusz Piwowar 

Academic Editor

PLOS ONE